# FlowBotHD: History-Aware Diffuser Handling Ambiguities in Articulated Objects Manipulation

**Yishu Li\*, Wen Hui Leng\*, Yiming Fang\*, Ben Eisner, David Held**
Robotics Institute, School of Computer Science
Carnegie Mellon University, United States
{yishul, wleng, yimingf, baeisner, dheld}@andrew.cmu.edu

**Abstract:** We introduce a novel approach for manipulating articulated objects which are visually ambiguous, such doors which are symmetric or which are heavily occluded. These ambiguities can cause uncertainty over different possible articulation modes: for instance, when the articulation direction (e.g. push, pull, slide) or location (e.g. left side, right side) of a fully closed door are uncertain, or when distinguishing features like the plane of the door are occluded due to the viewing angle. To tackle these challenges, we propose a history-aware diffusion network that can model multi-modal distributions over articulation modes for articulated objects; our method further uses observation history to distinguish between modes and make stable predictions under occlusions. Experiments and analysis demonstrate that our method achieves state-of-art performance on articulated object manipulation and dramatically improves performance for articulated objects containing visual ambiguities. Our project website is available at https://flowbothd.github.io/.

**Keywords:** Ambiguity, Articulated Objects, Diffusion

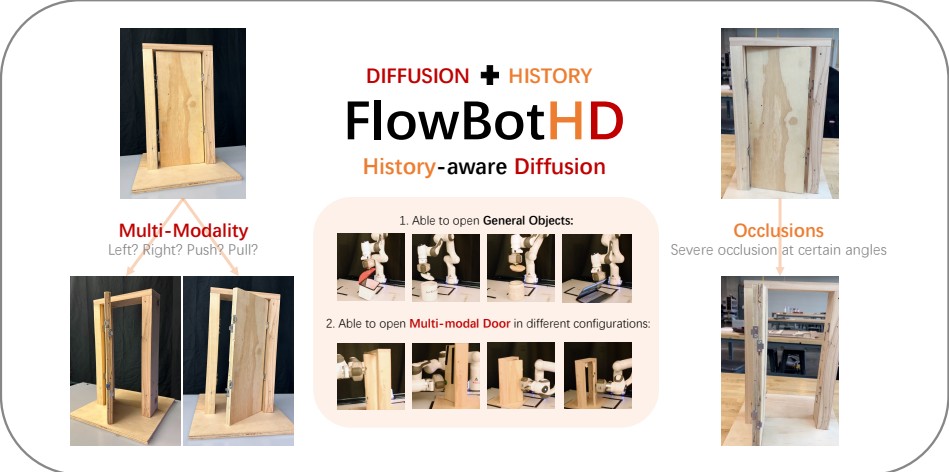

Figure 1: Our method, FlowBotHD, consists of a history-aware diffuser that handles multi-modality and occlusions in articulated object manipulation. Simulation and real world experiments show our model's capability of opening general objects (including handling ambiguities and occlusions) as well as an ambiguous door.

## 1 Introduction

Ambiguities are common in manipulation tasks. Consider opening a door without a visible handle or hinges: when the door is fully closed, the way in which it opens may be totally ambiguous.

8th Conference on Robot Learning (CoRL 2024), Munich, Germany.

It may swing open from the left side or the right side, and may swing either inwards or outwards. Without further information, even experienced humans can't always tell by sight whether it opens by push, pull or even slide. Another type of ambiguity is caused by occlusions. When a door opens to certain angles such that the door's principal plane is parallel to the viewing angle, it might look like a vertical line instead of a flat panel (see Figure 3). Occlusions cause loss of information, introducing ambiguities in the understanding of the object shape.

Faced with object ambiguities, a human might interact with the object and observe the outcome to disambiguate the true mode; for example, a person might push the door and see if it opens, and if not, they might try pulling the door open. Furthermore, momentary occlusions are not typically a problem for humans: we can remember past observations to supplement our current perception. We therefore would expect an intelligent robot to have the ability to learn from multi-modal training data, predict multi-modal solutions for ambiguous observations, and with the help of historical observations, make stable predictions even during occlusions.

To generalize across different articulated object kinematics and geometries, we build on prior work that models articulated object manipulation with 3D articulation flow (3DAF) vectors as proposed in Eisner et al. [1]. This approach predicts a set of 3D vectors for each point on the articulated object that indicates its opening direction. Results from this approach have shown impressive abilities to generalize to novel objects in the real world, after training only in simulation. However, since this prediction is unimodal and deterministic, this approach often fails to predict the correct flow when facing the ambiguous settings described above.

Therefore, in this paper, we propose a history-aware diffusion method to model multi-modality and handle occlusions for articulated objects. We use diffusion to model multi-modal distributions over multiple possible object opening directions, and we use historical observations to adjust or disambiguate the predicted distribution accordingly. Our experiments and analysis demonstrate the benefits of our method in such ambiguous situations. The contributions of this paper include:

(1) A novel approach to model multi-modal ambiguity of articulated objects caused by visual ambiguity or occlusion.
(2) A novel structure for incorporating object history to disambiguate articulation modes.
(3) Simulated experiments across a wide variety of PartNet-Mobility objects.
(4) Real-world experiments that demonstrate the feasibility of our method in real-world.

## 2 Related Work

**Articulated Object Manipulation**: Manipulating articulated objects is an important and challenging task. One line of work investigates building manipulation systems with analytical methods [2, 3, 4, 5, 6]. With the help of large-scale datasets [7, 8], recent lines of work turn to learning-based methods. Several works model manipulation choices with explicit articulation parameters and poses [9, 10, 11, 12]. To improve the model's generalization ability to different kinematics, several recent works learn to predict geometry-aware kinematics-free representations such as visual affordances [7, 13, 14] and articulation flows [1, 15] which achieve significant improvements in object generalization. Different from previous learning-based works which tend to use regression models for prediction, we propose to use diffusion models for better handling of multi-modality.

**Uncertainty Modeling in Robotics:** Modeling uncertainty is a well-explored topic in various robotics fields [16, 17]. Previous works have included uncertainty in tasks including visual perception and tracking [18, 19, 20, 21], motion planning [22, 23, 24, 25], safety control [26] and also articulation parameter estimation [27]. With similar motivation, our work focuses on building a model robust to visual uncertainty caused by multi-modality and occlusions.

**Diffusion Models in Robotics:** Diffusion models, a class of generative models, learns to denoise an input to approach a given distribution. Following its impressive success in image generation, diffusion has also shown great potential in robotics tasks. Recent works have applied diffusion models to various tasks of robotics, including reinforcement learning [28, 29], imitation learning [30,

31, 32, 33, 34] and motion planning [35, 36, 37]. In contrast to prior methods, we show how 3D diffusion models can be used to handle ambiguities in articulated object manipulation.

# 3 Background: 3D Articulation Flow (3DAF)

Our method builds upon the 3D Articulation Flow (3DAF) representation of Eisner et al. [1], which we briefly review here. We assume that the parts of the articulated object are connected in a kinematic tree with no closed-loop kinematic chains, and also that each joint has only one degree of freedom. Suppose that an articulated object is described by a point cloud $\mathbf{P} = \{p_0, \ldots, p_{N-1}\}$ for a set of 3D points $p_i \in \mathbb{R}^3$ on the object visible surface. Following Eisner et al. [1], for each point $p_i$, we define the 3D Articulation Flow (3DAF) $f_i \in \mathbb{R}^3$ as the motion that point would undergo if there were an infinitesimal positive displacement $\delta\theta$ of the joint between that point's articulation part and its parent. Eisner et al. [1] showed that, given a suction gripper, we can most efficiently open the articulated object using the predicted 3DAF by attaching the gripper to the point $p_{i_{\max}}$ with the largest predicted flow norm $i_{\max} = \arg\max_i \|f_i\|$ and then moving the gripper along its predicted flow direction. The set of 3D Articulation Flows $\mathbf{F} = \{f_0, \ldots, f_{N-1}\}$ for all points in the point cloud is hierarchy-free and invariant under object translation and scaling, which allows for effective learning over a large set of objects of different categories and effective transfer from simulation to the real world.

# 4 Method

The task we propose to tackle is opening fully-closed articulated objects from a single-view depth-sensor. The inputs to our model are a history of 3D point cloud observations of the articulated object being manipulated. We assume that the articulated objects do not consist of a closed-loop kinematic chain and that each joint has only a single degree of freedom.

We aim to enable a robot to manipulate ambiguous articulated objects, focusing on two types of ambiguity: ambiguity in opening direction due to lack of visual cues, and ambiguity due to self-occlusion during articulation. Both types of ambiguity are known failure modes of prior deterministic systems [1]. At a high level, our objective is to predict all the modes that describe how a potentially-ambiguous articulated object might open, and then to use these predictions to determine how to manipulate the object. Below we will describe our proposed history-aware diffusion model which can represent multimodality (Section 4.1) and is robust to occlusion (Section 4.2), as well as a system for manipulating articulated objects based on multimodal predictions (Section 4.3).

## 4.1 Diffusion for Multi-Modality

Our system builds on the 3D Articulation Flow representation from prior work [1] described in Section 3. Our first task is to construct a system which can accurately predict multi-modal affordances for an articulated object, in which affordances are captured by 3D Articulation Flow. For instance, given an ambiguous door that can open right or left (as shown in Figure 4), we'd like to be able to generate one set of flow vectors to describe the possibility that the door opens rightward, and a different set of flow vectors to capture the possibility that it might open leftward. Diffusion processes have been shown to be a good choice of model architecture for this setting, as they can represent multi-modal distributions over high-dimensional data [38, 30].

We now describe how we can use diffusion to predict multi-modal 3D Articulation flow. To incorporate observed 3D point cloud inputs $\mathbf{P}_{\text{obs}}$ into the standard diffusion model paradigm, we use a PointNet++ [39] encoder $g_\phi$ that encodes the current inputs and a DiffusionTransformer (DiT) [40] denoiser $\mathbf{D}_\theta$. During training, we first add noise to the ground truth flow $\mathbf{F}^*_{\text{obs}}$ to obtain a noisy flow $\mathbf{F}^\epsilon_{\text{obs}}$, following the DDPM [38] diffusion formulation. Then we pair each point and its corresponding noisy flow as the current state $\mathbf{S}_{\text{cur}} = \{p_{\text{obs},i}, f^\epsilon_{\text{obs},i}\}, i \in [1 \ldots N]$, where $N$ is the total number of observed points in $\mathbf{P}_{\text{obs}}$. We encode the current state $\mathbf{S}_{\text{cur}}$ as $\Psi_{\text{obs}} = g_\phi(\mathbf{S}_{\text{cur}})$ to obtain 3D geometry-aware point-wise features. Following the standard diffusion training procedure [38],

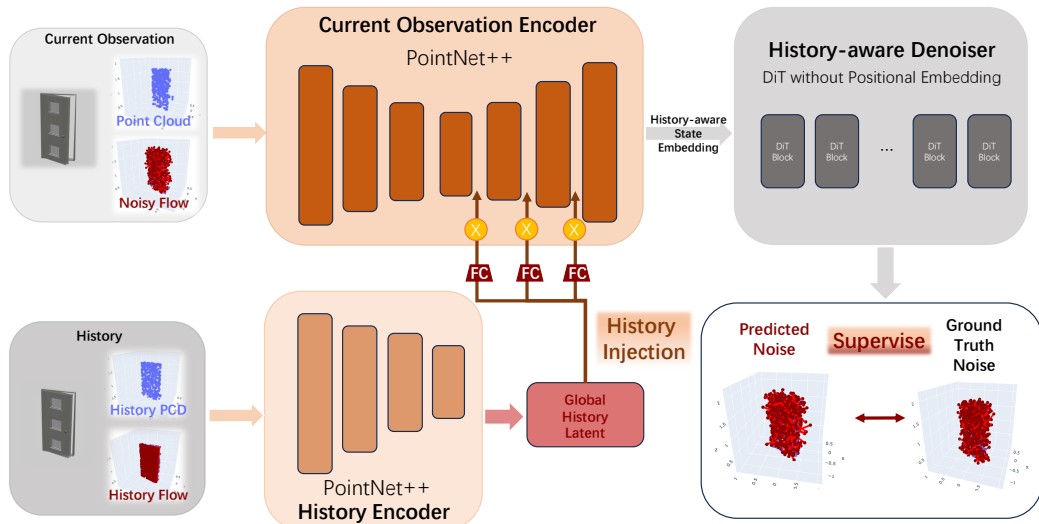

Figure 2: **FlowBotHD** structure: a history observation and the history flow are input into a history encoder to obtain a global history latent. This history latent is injected into the current observation encoder through fully-connected projection layers and element-wise multiplication. The current observation encoder outputs a history-aware point-wise embedding. The history-aware embedding is then input to a DiT-based denoiser to predict the 3D articulation flow [1], i.e. the predicted future motion of the object if it were to be opened a small amount.

we train the denoiser $\mathbf{D}_\theta$ and encoder $g_\phi$ with the following learning objective:

$$\mathcal{J}(\theta, \phi) = \mathbb{E}[||\epsilon_t - \mathbf{D}_\theta(\sqrt{\bar{\alpha}_t}x_0 + \sqrt{1 - \bar{\alpha}_t}\epsilon_t, t, g_\phi(\mathbf{S}_{\text{cur}}))||_2^2]$$

From the predicted noise, we estimate the flow $\hat{\mathbf{F}}$ following the standard diffusion equations [38].

## 4.2 Aggregating History for Disambiguation and Occlusion Handling

While this proposed 3DAF diffusion model is capable of modeling multi-modal distributions, it lacks the ability to disambiguate based on past interactions. For example, suppose the door is occluded in a current observation, as shown in Figure 3; if the robot succeeded in the past timesteps to partially open the door, it should output a distribution that places more probability on a flow which agrees with the previously observed motion.

To accomplish this objective, we propose to modify the above architecture to integrate the past history with a history encoder. We define a compact history state representation as the previous time step's point cloud observation $\mathbf{P}_{\text{his}}$, along with the previous step's flow $\mathbf{F}_{\text{his}}$, as $\mathbf{S}_{\text{his}} = \{p_{\text{his},i}, f_{\text{his},i}\}, i \in [1 \ldots M]$, where $M$ is the number of points in the previous point cloud. We then pass $\mathbf{S}_{\text{his}}$ though a global point cloud encoder $h_\gamma$ based on PointNet++ [39] to get a global history context vector $z_{\text{his}} = h_\gamma(\mathbf{S}_{\text{his}}) \in \mathbb{R}^{d_{\text{his}}}$, where $d_{\text{his}}$ is the fixed dimension of the history latent. We then inject this additional context into the "decoding" layers of our U-net encoder $g_\phi$, similar to the method proposed by Shridhar et al. [41] (see Figure 2). For each decoding layer $j$ of $g_\phi$ with point-wise activations $a_j \in \mathbb{R}^{d_j}$, we apply a learned linear projection $\mathbf{W}_j \in \mathbb{R}^{d_{\text{his}} \times d_j}$ to the history latent $z_{\text{his}}$, and combine with the original activation using the Hadamard product: $a'_j = a_j \odot (\mathbf{W}_j \cdot z_{\text{his}})$.

During training, we generate a synthetic history by sampling a random small angle to perturb the object to generate the "history" point cloud observation, and we obtain the corresponding ground truth articulation flow as the history flow. At inference time, we use the model's previous point cloud observations and its previous flow predictions as history. To account for situations at inference time when no history yet exists (i.e. the first timestep), we also train with samples that include no history, and include a learnable "start" vector $z_0$ in place of the history vector $z_{\text{his}}$ in such cases. We train this model end-to-end with the standard diffusion objective outlined in Section 4.1.

### 4.3  Articulation Manipulation Policy

Along with the history-aware diffusion model, we propose a novel articulation manipulation policy that utilizes the flow predicting model in real manipulation tasks, taking into consideration the multi-modality of diffusion and the incorporation of object history.

**Switch Grasp Point (SG):**  Following Eisner et al. [1], to achieve the best manipulation efficiency with a suction gripper, we want to grasp at the point with the greatest leverage; thus the optimal action given the predicted flow is to grasp the point that has the maximum flow norm [1]. Due to the multi-modality of diffusion, the prediction at different steps will vary; each prediction might imply a different point that the robot should grasp for maximum leverage. In order to correct for past potential grasping mistakes as well as to maintain a reasonable re-grasping cost, we propose to switch the grasp point whenever the leverage (defined as the current largest flow norm) of the newly predicted grasp point is larger than that of the current grasp point by a threshold: $||\hat{\mathbf{F}}_{max}|| - ||\hat{\mathbf{F}}_g|| > \epsilon_l$, where $\hat{\mathbf{F}}$ is the current step's flow prediction for all points in the point cloud, $g$ is the index of the point that is closest to the current grasp point, and $\epsilon_l$ is the threshold which we set to $0.2$ in practice.

**Consistency Check (CC):**  Learned diffusion models may not always perfectly represent the true distribution over history-conditioned 3DAF values; for example, they might predict multiple modes even when there is unambiguously only a single mode. To handle this, we perform a consistency check for every prediction after the first. Assuming that our model will sample the true mode given enough samples for any observation, we design a heuristic consistency check to filter the actions that are not consistent with the previously executed motion. Specifically, we define the consistency direction $g_{\text{his}}$ as the measured gripper motion caused by executing previous selected flow $\hat{f}_{\text{his}}$ (we only keep the previous execution if its motion was greater than $0$). For a subsequent step to be considered "consistent", the predicted action's direction shouldn't deviate too much from that of the previous step; thus we set an angle threshold $\epsilon_\theta$ for the consistency check: $\arccos \frac{\hat{f}_{\text{max}} \cdot g_{\text{his}}}{||\hat{f}_{\text{max}}|| \cdot ||g_{\text{his}}||} < \epsilon_\theta$, where $\hat{f}_{\text{max}}$ is the current maximum sampled flow direction $\hat{f}_{\text{max}} = \max_i \hat{f}_i$. We sample 3DAF flows from the diffusion model and compute maximum-leverage actions [1] until we find one that satisfies the consistency check, before executing the sampled action.

**History Filter (HF):**  If we rely solely on a consistency check for next actions, the model may end up in a degenerate state where its previous predictions are incorrect, enforcing that only incorrect predictions are permitted by consistency. Therefore, we include a history filter that only updates the history point cloud $\mathbf{P}_{\text{his}}$ and history flow $\mathbf{F}_{\text{his}}$ if the previous action successfully achieves reasonable gripper motion without losing contact. Specifically, after an execution, we only update the history if two criteria are satisfied:

1. The measured gripper motion $g_{\text{cur}}$ caused by the currently-executed action is consistent with past executions: $\arccos \frac{g_{\text{cur}} \cdot g_{\text{his}}}{||g_{\text{cur}}|| \cdot ||g_{\text{his}}||} < \epsilon_c = 90°$ where $g_{\text{cur}}$ is the current gripper movement and $g_{\text{his}}$ is the gripper movement from the previous execution.
2. The gripper motion $g_{\text{cur}}$ is larger than a threshold $||g_{\text{cur}}|| > \epsilon_m$ where $g_{\text{cur}}$ is the current gripper movement and $\epsilon_m$ is the threshold

The pseudocode for our method, FlowBotHD (History and Diffusion) can be found in Algorithm 1.

## 5  Experiments and Analysis

We use the PartNet-Mobility dataset [7] to train and evaluate the models. We split a subset of this dataset containing 21 categories of objects (the same categories as in prior work [13, 1]) into a train set and test set in which the test set consists of unseen samples of the training categories (707 training objects and 166 testing objects). To enable the models to open objects from a fully closed state, we create a mixed dataset which contains both randomly opened samples (as in prior work [1]) and training samples that are fully closed. The fully closed examples are the most ambiguous, since it might be unclear which direction these objects open.

**Algorithm 1:** The FlowBotHD articulation manipulation policy

---

**Require:** $\theta \leftarrow$ parameters of a trained FlowBotHD network $D$
  **while** `NotFullyOpened()` **do**
    $\mathbf{P}_{\text{obs}} \leftarrow$ Observation
    $\hat{\mathbf{F}} \sim D_\theta(\mathbf{P}_{\text{obs}}, \mathbf{S}_{\text{his}})$  // Sample the current flow direction from the diffusion model
    $\hat{f}_{\max} \leftarrow$ `GetMaxFlow`$(\hat{\mathbf{F}})$   // Find the point with the max predicted flow [1]
    **if** `Inconsistent`$(\hat{f}_{\max}, g_{\text{his}})$ and not `ExceedMaxTrialPerStep()` **then**
      continue ;            // If the sampled action fails the Consistency Check
    **end if**;               // (Sec. 4.3) then resample
    $p_{\text{grasp}} \leftarrow$ `SelectGraspPoint`$(\hat{\mathbf{F}})$
    **if** `NeedSwitchGraspPoint`$(p_{\text{grasp,his}}, p_{\text{grasp}})$ **then**
      $p_{\text{grasp,his}} = p_{\text{grasp}},$ ;        // If the Switch Grasp Point check is passed
      `Grasp`$(p_{\text{grasp}})$ ;         // (Sec. 4.3) then change the grasp point
    **end if**
    Apply a force to the gripper in the direction of $\hat{f}$ for small duration $\delta t$.
    $g_{\text{cur}} \leftarrow$ `GetDeltaGripper()`
    **if** `Consistent`$(g_{\text{cur}}, g_{\text{his}})$ **then**
      $g_{\text{his}} \leftarrow g_{\text{cur}}$
      **if** $||g_{\text{cur}}|| > \epsilon_m$ **then**
        $\mathbf{S}_{\text{his}} \leftarrow (\mathbf{P}_{\text{obs}}, \hat{\mathbf{F}})$ ;            // Apply the History Filter (Sec. 4.3)
      **end if**
    **end if**
  **end while**

---

## 5.1 Simulation results

To evaluate each model's ability to open objects, we construct a simulation environment in the Py-Bullet [42] simulator consisting of a suction gripper and objects from the PartNet-Mobility dataset, where the environment's goal is to successfully articulate the object. We initialize every object as fully closed and filter out instances that cannot open from the fully closed state even with the ground truth action sequence (e.g. due to self-collision). Following prior work [1], we compute 2 evaluation metrics: **Success Rate**: the percent of objects the model succeeds to open $> 90\%$ of the joint's full range of motion, and **Normalized Distance** [13]: the mean distance between the end state of this trial and the fully opened state, normalized with its range of motion $d_{\text{goal}} = \frac{||j_{\text{end}} - j_{\text{closed}}||}{||j_{\text{opened}} - j_{\text{closed}}||}$ where $j_{\text{end}}$ is the joint angle at the end of the policy rollout, $j_{\text{closed}}$ and $j_{\text{opened}}$ are the fully closed angle and the fully opened angle respectively. We evaluate each test object for 5 trials.

**Baselines:** To compare our model with the original FlowBot3D [1] setting, we trained FlowBot3D on a dataset with only randomly opened ("RO") samples which aligns with the original training strategy of Eisner et al. [1]. We also retrain Flowbot on the mixed dataset ("MD") that we use to train our model in which we also include examples in which the object is fully closed. Further, for a fair comparison to our method, we implement a baseline in which we allow FlowBot to switch its grasp point ("SG") using the same criteria as for our method.

Comparing the **Baseline** models with **Ours** in Tables 1 and 3 (see Appendix for normalized distance metrics), we can see that our full method (FlowBotHD) with the complete set of policy improvements (switch grasp point + consistency check + history filter) achieves the best overall performance. Specifically, compared to the original FlowBot3D [1] model, it increases the success rate averaged across all samples from 87% to 95%, or a 61% reduction in error rate (from 13% to 5%). We observe a particular increase in success rate in the door category (from 23% to 77%), which contains the most visual ambiguities of any category when instances are fully closed. These quantitative results demonstrate our model's ability to overall improve the performance and more importantly, to better handle multi-modal cases.

| | SG | CC | HF | $AVG_c$ | $AVG_s$ | | | | | | | | | | | | | | | | | | | | |
|---|---|---|---|---|---|---|---|---|---|---|---|---|---|---|---|---|---|---|---|---|---|---|---|---|---|
| **Baselines** | | | | | | | | | | | | | | | | | | | | | | | | | |
| FlowBot (RO) | × | × | × | 0.800 | 0.865 | **0.85** | 0.50 | 0.75 | 0.71 | **1.00** | 0.30 | 0.91 | 0.80 | 0.75 | 0.69 | 0.89 | 0.90 | **0.89** | 0.96 | 1.00 | 1.00 | 0.23 | 1.00 | 0.86 | 1.00 |
| FlowBot (RO) | ✓ | × | × | 0.793 | 0.903 | **0.85** | 0.50 | 0.75 | 0.82 | 1.00 | 0.60 | 0.87 | 0.80 | 0.58 | 0.88 | 0.96 | 0.94 | 0.82 | **0.96** | 0.73 | 0.60 | 0.53 | 1.00 | 0.86 | 0.80 |
| FlowBot (MD) | × | × | × | 0.800 | 0.875 | 0.60 | 0.80 | **1.00** | 0.82 | 1.00 | 1.00 | 0.89 | 0.76 | 0.33 | 0.72 | 0.91 | 0.91 | 0.78 | 0.92 | 0.80 | 0.40 | 0.37 | 1.00 | 1.00 | 1.00 |
| FlowBot (MD) | ✓ | × | × | 0.842 | 0.926 | 0.75 | **1.00** | **1.00** | 0.96 | 1.00 | 1.00 | 0.95 | 0.80 | 0.33 | **0.99** | 0.95 | 0.95 | **0.89** | 0.96 | 0.73 | 0.40 | 0.40 | 1.00 | 0.89 | 0.90 |
| **Ablations** | | | | | | | | | | | | | | | | | | | | | | | | | |
| Ours- No History | × | × | × | 0.692 | 0.806 | 0.70 | 0.00 | 0.75 | 0.84 | 0.84 | 0.60 | 0.82 | 0.72 | 0.44 | 0.59 | 0.80 | 0.86 | 0.76 | 0.72 | 0.93 | 0.20 | 0.47 | 1.00 | 0.86 | 0.95 |
| Ours- No History | ✓ | × | × | 0.740 | 0.864 | 0.70 | 0.40 | 0.85 | 0.93 | **1.00** | 1.00 | 0.76 | 0.80 | 0.33 | 0.69 | 0.91 | 0.90 | **0.89** | 0.80 | 0.73 | 0.00 | 0.53 | 1.00 | 0.86 | 0.70 |
| Ours- No History | ✓ | ✓ | × | 0.743 | 0.876 | 0.70 | 0.30 | 0.75 | 0.95 | 0.92 | 1.00 | 0.73 | 0.80 | 0.11 | 0.84 | 0.94 | 0.91 | 0.84 | 0.92 | 0.87 | 0.20 | 0.53 | 1.00 | 0.86 | 0.70 |
| Ours- No Diffusion | ✓ | × | × | 0.673 | 0.720 | 0.55 | 0.30 | 0.45 | 0.89 | 1.00 | 0.40 | 0.85 | 0.80 | 0.28 | 0.56 | 0.82 | 0.70 | 0.58 | 0.16 | 0.87 | 1.00 | 0.40 | 1.00 | 0.86 | 1.00 |
| Ours- No Diffusion | ✓ | × | ✓ | 0.683 | 0.735 | 0.50 | 0.30 | 0.55 | 0.89 | 1.00 | 0.10 | 0.82 | 0.80 | **0.97** | 0.56 | 0.86 | 0.72 | 0.58 | 0.08 | 0.60 | 1.00 | 0.53 | 1.00 | 0.80 | 1.00 |
| **Ours** | | | | | | | | | | | | | | | | | | | | | | | | | |
| FlowBotHD | ✓ | × | × | 0.735 | 0.865 | 0.35 | 0.10 | 0.90 | **1.00** | 0.96 | 0.50 | 0.89 | 0.80 | 0.36 | **0.99** | 0.85 | 0.92 | 0.49 | 0.92 | 0.47 | **1.00** | 0.43 | 1.00 | 0.83 | 0.95 |
| FlowBotHD | ✓ | ✓ | × | **0.884** | 0.937 | 0.65 | 0.70 | 0.95 | **1.00** | 1.00 | 1.00 | 0.96 | 0.96 | 0.82 | 0.97 | 0.94 | 0.96 | 0.64 | **0.96** | 0.73 | 0.80 | 0.70 | 1.00 | 0.97 | 0.95 |
| FlowBotHD | ✓ | × | ✓ | 0.758 | 0.904 | 0.30 | 0.00 | 0.85 | **1.00** | 0.96 | 0.90 | **0.96** | 0.80 | 0.45 | **0.99** | 0.95 | 0.95 | 0.64 | 0.92 | 0.53 | 0.80 | 0.43 | 1.00 | 0.86 | 0.85 |
| **FlowBotHD** | ✓ | ✓ | ✓ | 0.878 | **0.952** | 0.75 | 0.70 | 0.95 | 0.98 | **1.00** | 0.70 | **0.96** | 0.84 | 0.58 | **0.99** | 0.98 | 0.98 | 0.73 | **0.96** | 0.87 | **1.00** | 0.77 | 1.00 | 0.83 | 1.00 |

Table 1: Success Rate Metric Results (↑): Fraction of successful trials (normalized distance < 0.1) of different object categories after a full rollout across different methods; the higher the better. $\mathbf{AVG}_c$ and $\mathbf{AVG}_s$ refers to metric averaged over categories or samples. For the policy column, $SG$ refers to switch grasp point, $CC$ refers to consistency check and $HF$ refers to history filter. We only add history filter in history-aware methods and only add consistency check with diffusion-based methods because inconsistent predictions in deterministic methods can't be refined even with re-predictions.

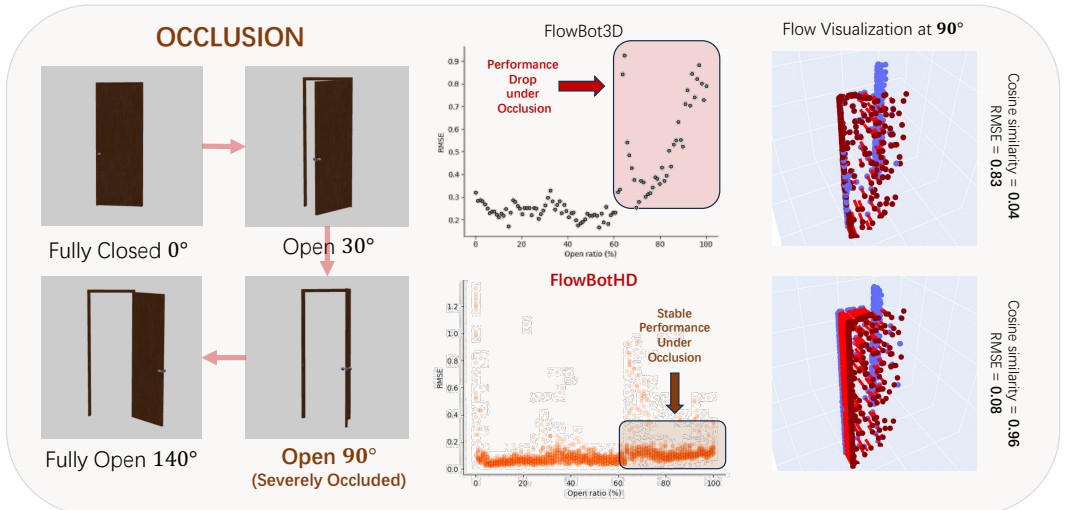

Figure 3: We demonstrate our model's performance improvement over the baseline in a severely occluded case. We open the object to different angles, make predictions, and plot the RMSE metric (↓) against the open ratio. The baseline (top) fails at the point of severe occlusion, whereas our method (bottom) continues to make stable predictions using history. We also include flow visualizations (right) to show the quality of the predictions in both cases.

## 5.2 Ablations

To understand the impact of our design decisions, we perfomed ablation studies in which we remove various components of the model: No history, no diffusion, and varying whether we use switch grasping ("SG" column), consistency check ("CC" column), and history filtering ("HF" column). Looking at Tables 1 and 3, we observe large performance improvements when including history awareness in the diffusion model (comparing Ours to "Ours- No History"). This improvement demonstrates that the extra guidance of history is beneficial for diffusion to reduce the multimodality in cases of past success in which the opening direction is now disambiguated.

These tables show that our policy improvements of switching grasp point, applying a consistency check, and applying a history filter all contribute to increased model performance. The importance of the consistency check potentially implies that one challenge for using diffusion models for visual

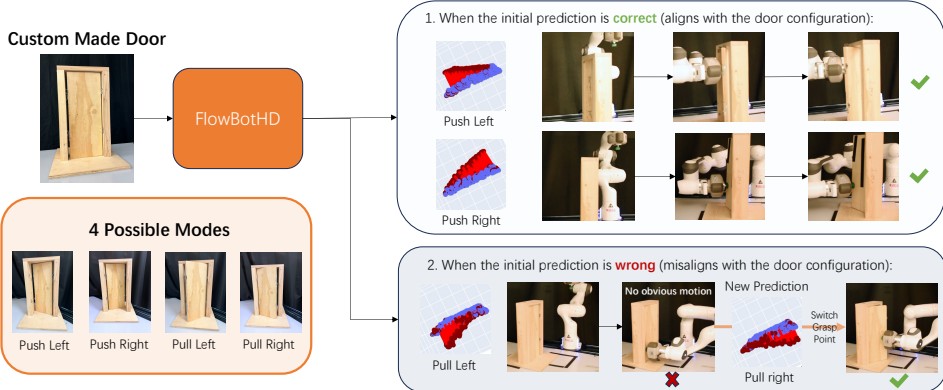

Figure 4: Demonstration of FlowBotHD opening a custom made door in different configurations. The first two rows show examples in which the model's initial predictions aligned with the current door configuration. The last row shows an example in which the model's initial prediction fails and the model switches its grasp point, ultimately leading to a success.

manipulation policies is randomness and, as a result, inconsistency. By filtering out inconsistent predictions, our policy takes actions that lead to higher success rates than without filtering.

## 5.3 Qualitative Analysis

To understand the ability of our model to handle occlusions which are not immediately apparent from our quantitative metrics, in Figure 3, we visualize the model predictions for different door opening angles. At certain angles, the point cloud is severely occluded such that only the vertical edge of the door is visible from the camera's perspective. We can see from the RMSE plot that FlowBot3D consistently mispredicts the 3DAF on examples where the opening angle causes severe occlusions. FlowBotHD performs substantially better in these occluded cases by leveraging history, predicting the correct direction even at the point of occlusion. The flow visualization shows a clear improvement between our model and the baseline for this ambiguous case.

## 5.4 Real-World Experiments

We performed real-world experiments on general articulated objects, as well as a custom made multi-modal door to demonstrate FlowBotHD's ability to function in real-world ambiguous scenarios (Figures 1 and 4). For the multi-modal door, we demonstrate our model's ability to open the door in all 4 possible modes, and also the ability to adjust to better grasp points with the switch grasp point policy. Static visualizations of the real-world roll-outs on our custom-made door and general articulated objects are presented in Figure 4 and Appendix Figure 15, and full videos can be found on our project website. Details of our real-world setup can be found in Appendix F.

## 6 Conclusion

In this work, we propose a history-aware diffusion model **FlowBotHD** to handle ambiguities encountered while manipulating articulated objects, along with a carefully designed heuristic manipulation policy. Both quantitative experiments and qualitative analysis show the overall effectiveness of our model and specific improvements in multi-modal or severely occluded cases.

**Limitations:** While our model greatly improves the ability of the policy to handle ambiguities, we have observed trade-offs in accuracy and consistency when building on top of a diffusion pipeline. Our diffusion model sometimes makes multimodal predictions even when the object is opened enough to disambiguate the true opening direction. While we currently mitigate this issue by adding a consistency check in the policy, an ideal model would learn to disambiguate perfectly based solely history. Therefore, we hope to further explore how to exert better control over the diffusion's learned distributions for better consistency and accuracy.

**Acknowledgments**

This material is based upon work supported by the National Science Foundation under NSF CA-REER Grant No. IIS-2046491 and Toyota Research Institute.

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

# Appendix

## Table of Contents

## A   Training Details

**Model Structure:** FlowBotHD consists of a current state encoder $g_\phi$, a history state encoder $h_\gamma$ and a denoiser $\mathbf{D}_\theta$. Both current and history state encoder are based on PointNet++ [39], where the history encoder only includes the PointNet++ global encoding module that outputs a global latent of 128 dimensions. We kept the same model architecture and hyperparameter set as the original PointNet++ paper. We inject the history latent into every layer of the PointNet++ framework in the decoder (see Figure 2). This is performed through a Hadamard product of the current point cloud encoding with the latent encodings at every decoding layer to produce the final output of the model. We base the denoiser on a DiT with 5 layers, 4 heads, and a hidden size of 128.

**Dataset Details:** Similar to the original FlowBot3D paper [1], we create a randomly opened dataset (RO) in which we randomly sample the view perspective, the target joint, and the open ratio. To open objects from the fully closed state, we create another mixed dataset (MD) in which we half of the objects are fully closed and the other half randomly opened. To enable history-aware training, we generate a dataset with 1/3 of the objects fully closed, 1/3 randomly opened but without history and the final third randomly opened and with history at random interval.

**Training Process:** We set diffusion timestep to be 100 steps. We train the model for 450 epochs using AdamW optimizer with a weight decay of 1e-5. We use a learning rate of 1e-4, and a batch size of 128. We evaluate the model every 20 epochs, and save the checkpoint with the best winner-take-all RMSE metric. The Winner-take-all metric means that we repeatedly make predictions for each sample 20 times and record the best RMSE.

# B    Simulation Details

We design a simulation environment in the PyBullet simulator that simulates operating a suction gripper to operate PartNet-Mobility objects. We first roll out the simulation with ground truth flow and filter out samples that can't open with the ground truth motion, due to errors in the simulator. We repeat each sample for 5 times during evaluation. The final test object categories we evaluated on after filtering and repeating are listed in Table 2.

| Icon | Name | Count | Icon | Name | Count | Icon | Name | Count | Icon | Name | Count |
|---|---|---|---|---|---|---|---|---|---|---|---|
|  | Oven | 20 |  | Microwave | 10 |  | Table | 245 |  | Phone | 5 |
|  | Bucket | 10 |  | Window | 55 |  | Furniture | 765 |  | Door | 30 |
|  | Box | 20 |  | Kettle | 25 |  | Dishwasher | 45 |  | Laptop | 40 |
|  | Toilet | 55 |  | Safe | 12 |  | WashMachine | 25 |  | TrashCan | 35 |
|  | KitchenPot | 25 |  | Refrigerator | 75 |  | FoldingChair | 15 |  | Stapler | 20 |

Table 2: Simulation Objects and Sample Counts

To manipulate the objects, we first attach the suction gripper to the object. This is achieved by first teleporting the gripper to a position close to the target grasp point and then gradually moving towards the target grasp point until contact is detected. After contact, we apply a physical force constraint between the gripper and the object. The move action is then implemented as applying a certain velocity along the predicted direction.

**Hyperparameter in Policy:**  There are 3 hyperparameters to set in Algorithm 1: the threshold for switch grasp point $\epsilon_l$, the threshold for consisteny check $\epsilon_\theta$ and the threshold for good movement $\epsilon_m$. We set $\epsilon_l = 0.2$, $\epsilon_\theta = 30°$ and $\epsilon_m =$1e-2 in simulation. Table 3 demonstrates the quantitative simulation results for the normalized distance metric, computed in the same way as in Eisner et al. [1].

| | SG | CC | HF | $AVG_c$ | $AVG_s$ | Oven | Bucket | Box | Toilet | KitchenPot | Microwave | Window | Kettle | Safe | Refrigerator | Table | Furniture | Dishwasher | WashMachine | FoldingChair | Phone | Door | Laptop | TrashCan | Stapler |
|---|---|---|---|---|---|---|---|---|---|---|---|---|---|---|---|---|---|---|---|---|---|---|---|---|---|
| **Baselines** | | | | | | | | | | | | | | | | | | | | | | | | | |
| FlowBot (RO) | × | × | × | 0.250 | 0.166 | 0.23 | 0.54 | 0.29 | 0.29 | **0.00** | 0.90 | 0.10 | 0.23 | 0.35 | 0.31 | 0.10 | 0.15 | 0.18 | 0.10 | 0.10 | 0.48 | 0.39 | **0.07** | 0.11 | 0.08 |
| FlowBot (RO) | ✓ | × | × | 0.220 | 0.117 | **0.16** | 0.53 | 0.29 | 0.22 | **0.00** | 0.43 | 0.10 | 0.15 | 0.45 | 0.14 | 0.05 | 0.09 | 0.18 | 0.09 | 0.27 | 0.31 | 0.46 | **0.07** | 0.15 | 0.26 |
| FlowBot (MD) | × | × | × | 0.183 | 0.143 | 0.31 | **0.19** | 0.06 | 0.22 | **0.00** | **0.06** | 0.14 | 0.21 | 0.47 | 0.30 | 0.09 | 0.12 | 0.23 | 0.12 | 0.09 | 0.24 | 0.63 | **0.07** | **0.05** | 0.08 |
| FlowBot (MD) | ✓ | × | × | 0.159 | 0.098 | 0.26 | 0.08 | 0.07 | 0.09 | **0.00** | **0.06** | 0.08 | 0.14 | 0.56 | **0.07** | 0.05 | 0.09 | 0.16 | 0.09 | **0.07** | 0.35 | 0.60 | **0.07** | 0.13 | 0.16 |
| **Ablations** | | | | | | | | | | | | | | | | | | | | | | | | | |
| Ours- No History | × | × | × | 0.255 | 0.176 | 0.29 | 0.57 | 0.18 | 0.19 | 0.16 | 0.36 | 0.14 | 0.23 | 0.46 | 0.28 | 0.18 | 0.14 | 0.20 | 0.31 | **0.07** | 0.53 | 0.54 | **0.07** | 0.09 | 0.10 |
| Ours- No History | ✓ | × | × | 0.178 | 0.113 | 0.26 | 0.36 | 0.09 | 0.09 | **0.00** | 0.06 | 0.13 | 0.10 | 0.38 | 0.17 | 0.08 | 0.09 | 0.12 | 0.21 | 0.31 | 0.29 | 0.45 | **0.07** | 0.10 | 0.20 |
| Ours- No History | ✓ | ✓ | × | 0.176 | 0.110 | 0.28 | 0.28 | 0.19 | 0.10 | 0.08 | **0.06** | 0.10 | 0.12 | 0.75 | 0.11 | 0.07 | 0.10 | **0.10** | 0.08 | 0.10 | 0.11 | 0.45 | **0.07** | 0.11 | 0.27 |
| Ours- No Diffusion | ✓ | × | × | 0.265 | 0.213 | 0.35 | 0.67 | 0.55 | 0.15 | **0.00** | 0.51 | 0.14 | 0.19 | 0.73 | 0.31 | 0.15 | 0.20 | 0.31 | 0.37 | 0.11 | **0.08** | 0.55 | **0.07** | 0.12 | 0.08 |
| Ours- No Diffusion | ✓ | × | ✓ | 0.194 | 0.183 | 0.39 | 0.48 | 0.40 | 0.11 | **0.00** | 0.45 | 0.18 | 0.17 | **0.04** | 0.30 | 0.11 | 0.18 | 0.28 | 0.35 | 0.24 | **0.08** | 0.48 | **0.07** | 0.15 | **0.07** |
| **Ours** | | | | | | | | | | | | | | | | | | | | | | | | | |
| FlowBotHD | ✓ | × | × | 0.181 | 0.139 | 0.55 | **0.50** | 0.13 | 0.05 | 0.04 | 0.39 | 0.12 | 0.15 | 0.33 | 0.08 | 0.15 | 0.10 | 0.40 | 0.11 | 0.23 | 0.09 | 0.52 | **0.07** | 0.10 | 0.09 |
| FlowBotHD | ✓ | ✓ | × | 0.103 | 0.086 | 0.33 | 0.35 | 0.11 | **0.05** | **0.00** | **0.06** | 0.06 | **0.02** | 0.15 | 0.08 | 0.07 | 0.08 | 0.25 | 0.09 | 0.08 | 0.10 | 0.27 | **0.07** | 0.07 | 0.08 |
| FlowBotHD | ✓ | × | ✓ | 0.110 | 0.096 | 0.53 | 0.52 | 0.13 | **0.05** | 0.04 | 0.10 | **0.06** | 0.15 | 0.24 | **0.07** | 0.05 | 0.07 | 0.21 | 0.13 | 0.23 | 0.11 | 0.49 | **0.07** | 0.12 | 0.10 |
| **FlowBotHD** | ✓ | ✓ | ✓ | **0.096** | **0.072** | 0.25 | 0.26 | **0.05** | **0.05** | **0.00** | 0.20 | **0.06** | 0.11 | 0.38 | **0.07** | **0.04** | **0.06** | 0.18 | **0.07** | **0.07** | 0.09 | **0.23** | **0.07** | 0.09 | **0.07** |

Table 3: Normalized Distance Metric Results ($\downarrow$): Normalized distances to the goal articulation joint angle after a full rollout of the methods. The lower the better.

We demonstrate the simulation process in Fig 5. We visualize the flow predictions (the first row), the simulation process (the last row, x-axis is the step number, the y-axis is the open ratio), and the trajectory plot along with policy signals (the middle plot). In these visuals, we can see our model and policy's pattern of opening an object: make one or more attempts at opening at the beginning of the episode, and once the door is opened a bit, make consistent predictions based on past history and consistency check.

The first example is quite smooth: our model succeeds in moving the door on the first step. It continues to make consistent and history-aware predictions until the door is fully opened. Qualitatively, the consistency check didn't filter out many predictions due to high prediction quality, and due to the smooth execution the history is updated at each step, meaning that the policy always uesd the previous step's history for the next step's prediction. In the second example, our model makes several attempts to open the object when in the fully closed state. The observation history is not updated during these steps. Once the door is opened a bit, the history information and the consistency check enable the model to execute consistent actions. In the bar plot, we se that for step 11, the consistency check filters out 12 inconsistent predictions, demonstrating the effectiveness of applying a consistency check. Also, we can see that at step 6, the door is opened a bit, but the history is not updated due to the small movement (indicating the prediction is not good enough) which demonstrates the importance of tuning the history filter.

## C  Multi-modality Analysis

### C.1  Multi-modality of Dataset

Analyzing how many of the objects in the dataset are multi-modal or typical multi-modal paradigms provides valuable insights of the dataset. Take door category as an example, doors without handle typically have 4 possible modes when fully-closed (push left, push right, pull left and pull right); handles can indeed disambiguate which side of the door opens but still leaves ambiguity on whether the door opens by push or pull. Among the 28 training door joints and 7 test door joints, most of them (20 out 28 for training, 5 out of 7 for test) are with handles (2 possible modes - push and pull), and the other samples are without handles (4 possible modes - push left, pull left, push right and pull right).

For other object categories, since there is no generic way to determine whether an object is ambiguous, it is hard to manually count ambiguous samples for all the categories (not to mention ambiguity might be subjective), therefore we describe broadly where ambiguities typically happen. There are three typical multi-modal objects (See Figure 6) : 1) Joints without handles, 2) Joints with small handles that might be erased when downsampling the point cloud, and 3) Joints with handles that may afford different opening modes: for example a dishwasher/oven with a handle on the top of its front surface, although the handle eliminates many modes, it is unclear whether it opens by pulling out like a drawer, or pulling from the top to the bottom like a lid or door.

Similar to counting modes of general articulated objects, there is no precise way to count multi-modality in predictions without including manual labeling. To approximately evaluate the multi-modal predictions of the model, we make 50 predictions for each object in the test set and calculate their RMSE with respect to the ground truth flow. We record how many of the objects include both RMSE $< 0.2$ (presumably a correct prediction) and RMSE $> 0.6$ (presumably a different mode than the ground truth mode). We can see from the bar plot in Figure 7 that the model predicts multi-modality mostly in conceptually multi-modal categories like Door, WashingMachine, and Refrigerators. However it also makes multi-modal predictions with conceptually unambiguous cases, which aligns with limitations of this work we addressed in 6 about the difficulty of striking a balance between precision and multi-modality.

### C.2  Model Performance on Multi-modal Cases

To specifically analyze our model's improvements on handling multi-modality, we evaluated baseline FlowBot3D trained on randomly opened dataset (RO) / mixed dataset (MD) and our model FlowBotHD on doors. We open each test door from 0% to 100%. We average the metrics within 10% open to compute the performance for closed doors, and we average the metrics above 10% open to compute the performance for open doors. As we can see from Table 4, our model outperforms the baselines by a large margin across all metrics. We observe that FlowBot3D trained on randomly

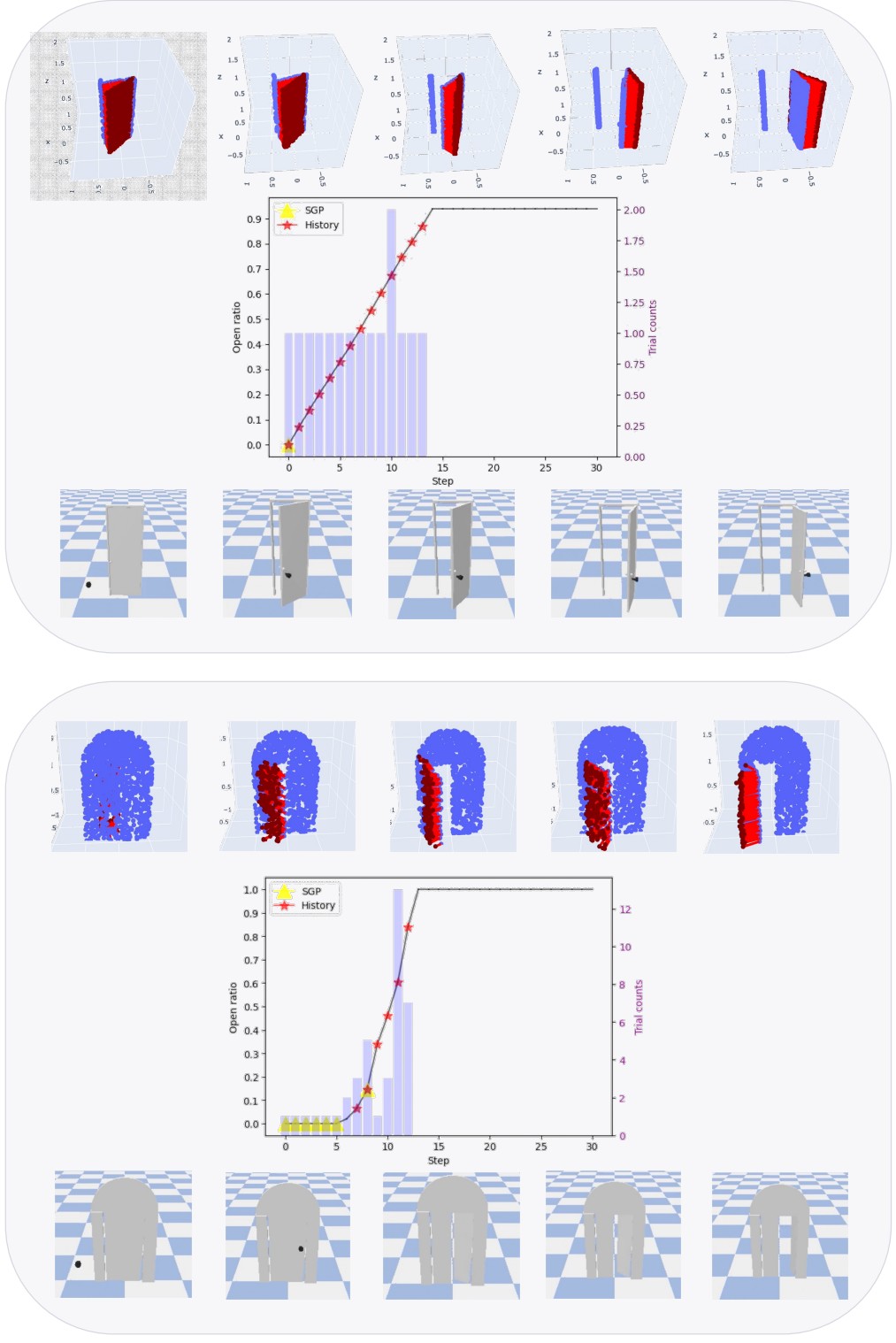

Figure 5: Simulation Visualizations: The plot in the middle is a simulation trajectory plot with x-axis as step number, and left y axis as the open ratio. We visualize the history update signal with red polygons, step with red polygon means updating this step's prediction as the latest history. Yellow triangle represents the switch grasp point (SGP) signal, meaning that this step requires a new grasp point. The bar plot on the background corresponds to number of trials we take to generate a prediction that satisfies the consistency check trial. The axis for the bar plot is the right y-axis.

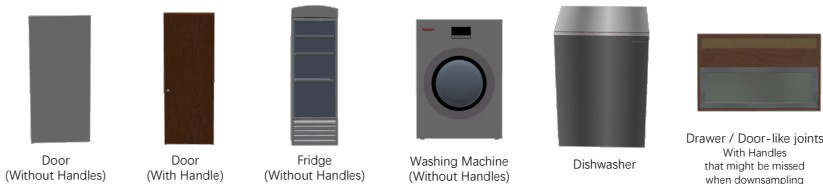

Door
(Without Handles)

Door
(With Handle)

Fridge
(Without Handles)

Washing Machine
(Without Handles)

Dishwasher

Drawer / Door-like joints
With Handles
that might be missed
when downsampling

Figure 6: Ambiguous Showcases: Conceptually, ambiguities typically happen with 1) Joint without handles, 2) Joints with small handles that might be missed when downsampling, and 3) Joints with handles which are still possible to open in different ways.

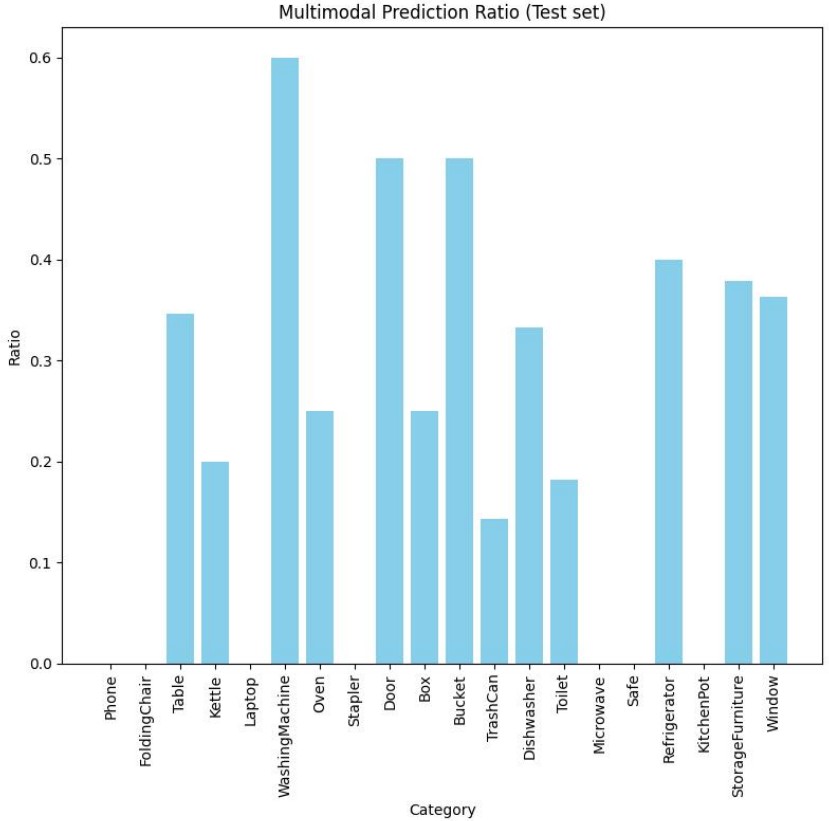

Figure 7: Multi-modal Prediction Ratio for Different Categories: We make 50 predictions for each object in the test set, calculate their RMSE with the ground truth flow, and record how many of the objects include both correct prediction and different modes.

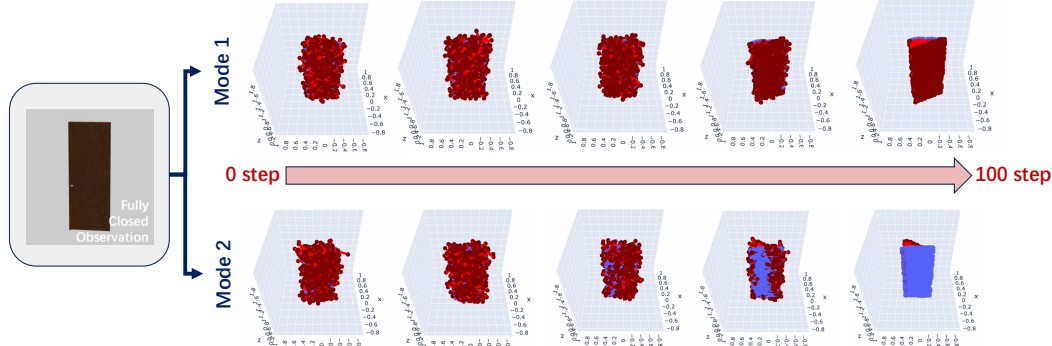

Figure 8: Multi-Model Diffusion Process Visualization: We visualize the denoising process (100 steps). We can see that the flow starts from pure random noises and gradually converges to different modes: Pull and Push.

opened dataset outperforms the one trained on the mixed dataset, showing that multi-modal training data can confuse the regression models' training process.

| Model | Cosine (↑) | | | RMSE(↓) | | | MAG(↓) | | |
|---|---|---|---|---|---|---|---|---|---|
| | FlowBot3D (RO) | FlowBot3D (MD) | **FlowBotHD** | FlowBot3D (RO) | FlowBot3D (MD) | **FlowBotHD** | FlowBot3D (RO) | FlowBot3D (MD) | **FlowBotHD** |
| Closed (<10% open) | 0.2033 | 0.3129 | **0.8046** | 0.4426 | 0.4069 | **0.1833** | 0.2468 | 0.2150 | **0.1047** |
| Randomly Open (>10% open) | 0.7351 | 0.2720 | **0.9089** | 0.2218 | 0.4617 | **0.1246** | 0.1782 | 0.2215 | **0.0919** |

Table 4: Multi-Modality Analysis: We compare our model with baselines on doors. The cosine metric means the cosine similarity between the predicted flow and the ground truth flow, the higher the better. RMSE metric means the root mean square error between prediction and ground truth, the lower the better. MAG metric means the flow magnitude error, the lower the better.

Fig 8 visualizes the denoising process that produces different modes based on the same fully closed door, demonstrating our model's ability of preserving multi-modality for ambiguous examples.

## D   Occlusion Analysis

We analyze our model's performance under occlusions on different object categories. A door example is included in the Fig 3 of the main paper, and we include occluded fridge and furniture examples in Fig 9. We can see from the visualizations that with history, FlowBotHD is able to produce stable and robust predictions regardless of the occlusions while FlowBot3D makes less consistent predictions when severely occluded.

## E   Policy Analysis

We conduct further analysis on components of our roll-out policies, including the occasional negative effect of the history filter, the efficiency of switching the grasp point during policy execution and the effect of limiting the number of steps during execution.

### E.1   History Filter Analysis

Since we train the model with only ground truth data, we propose to use a history filter to prevent incorrect predictions in the history from disturbing the history distribution; our history filter removes previous predictions that fail to further open the object. However, occasionally we find that including incorrect predictions in the history can lead to better performance; below we analyze when this might occur.

Consider the below dishwasher (which opens by pulling the upper side) as an example, we experimented with 4 types of history input: No history (which can occur either at the beginning or due

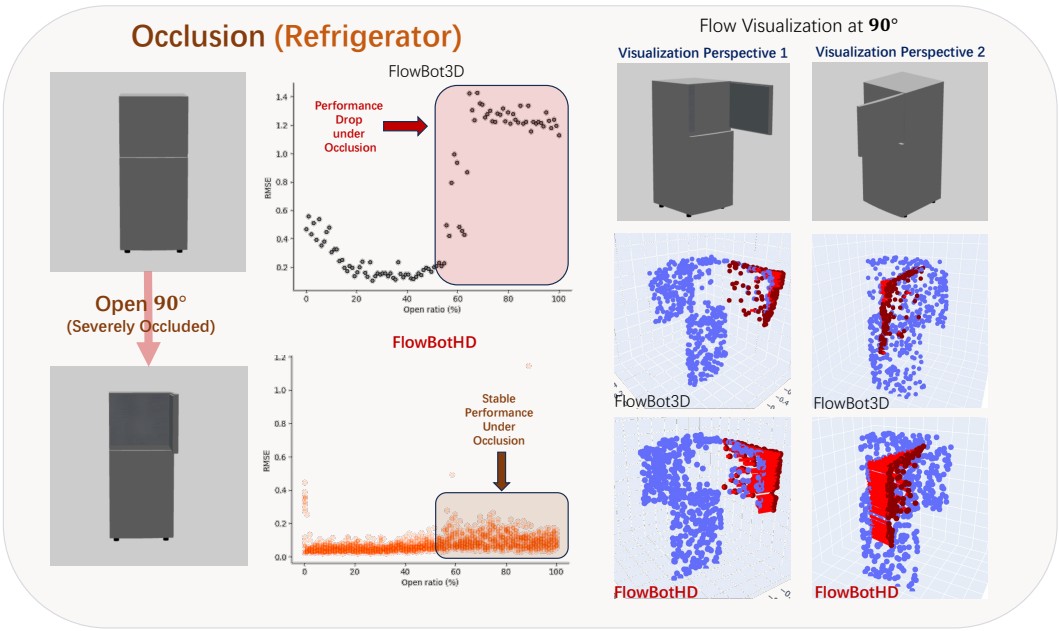

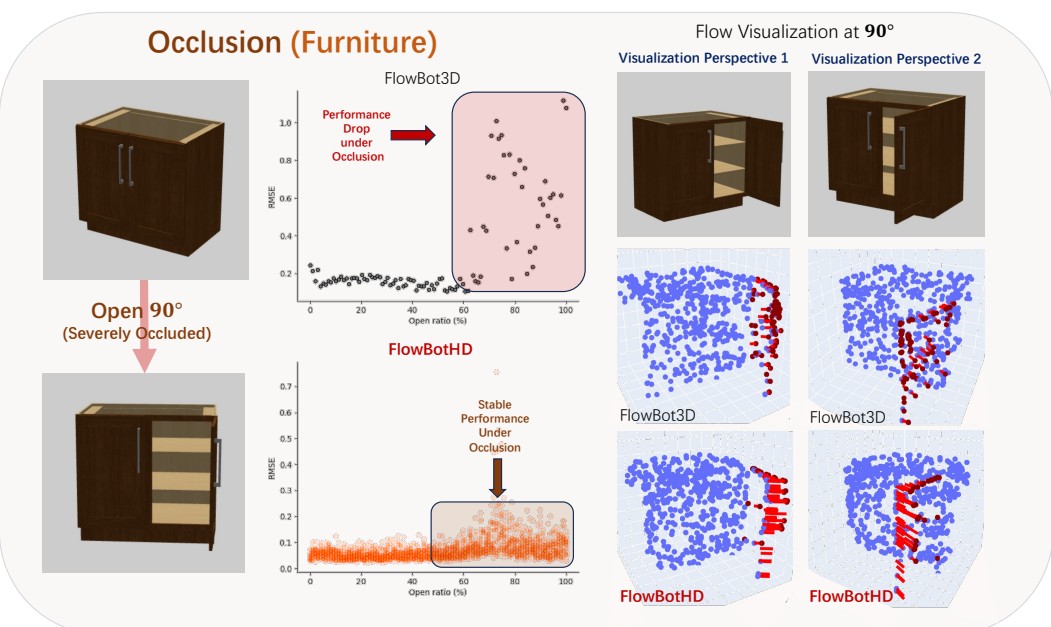

Figure 9: Occlusion analysis: We open the object to different angles, make predictions and plot the Cosine Similarity metric (↑) and RMSE metric (↓) against the open ratio. We also include flow visualizations from different viewing perspective to intuitively show the quality comparison of the predictions. We can see from the visualization that FlowBotHD can make good and consistent predictions even under severe occlusions.

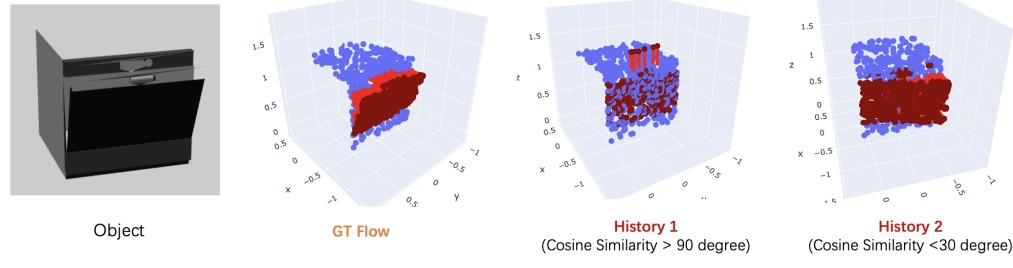

Object       GT Flow       History 1 (Cosine Similarity > 90 degree)       History 2 (Cosine Similarity <30 degree)

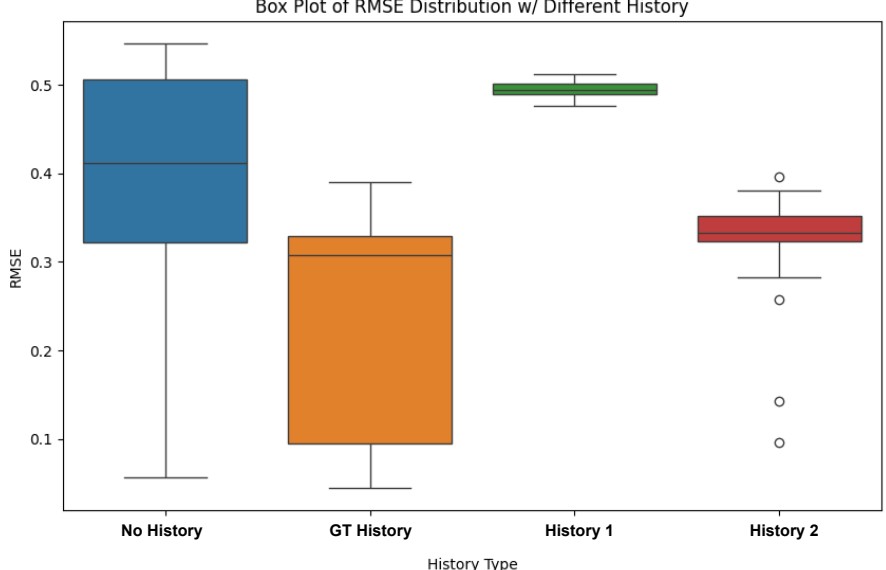

Figure 10: History Filter Analysis: Above shows the visualization of three types of history, the box plot below visualizes the effect of different histories on the next step's prediction quality.

to the history filter removing incorrect predictions), history from ground-truth predictions, history from predictions with cosine similarity to the ground-truth > 90 degrees, history from predictions with cosine similarity to the ground-truth < 30 degrees..

We can see from the box plot of RMSE distribution (the lower, the better) of 50 samples in Figure 10, that the history from the ground-truth predictions leads to the best performance on average, and the history from predictions with cosine similarity to the ground-truth > 90 degrees leads to worse performance than no history. However, interestingly, we also see that including history from predictions with cosine similarity to the ground truth < 90 degrees sometimes leads to better performance than including no history at all.

This experiment shows that including history from incorrect predictions can sometimes lead to better performance and sometimes worse depending on the accuracy of the previous predictions in the history. This implies that the history filter can sometimes inadvertently filter out helpful predictions in the history that might be close to the ground truth. Nonetheless, our results show that using the history filter on average leads to better performance than not using it.

## E.2 Switch Grasp Point Analysis

Switching grasp point costs extra time, therefore we should only allow necessary switches to happen. In our method, the stablility of grasp selection is achieved via the leverage change threshold $\delta l$ as proposed in Section 4.3. By adjusting this threshold, we can control how frequently the policy switches the grasp point.

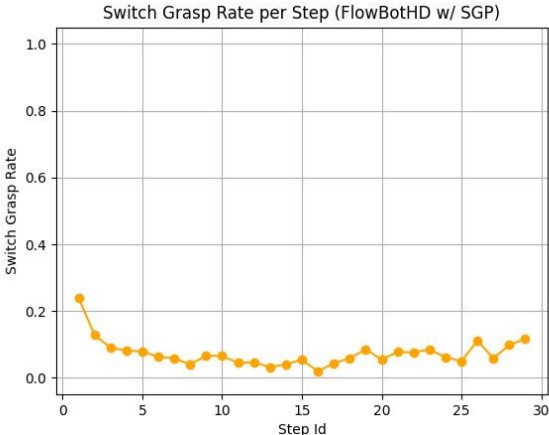

Figure 11: Switch Grasp Rate per Step Plot: As seen in the plot, the switch grasp point rate gets lower among the first few steps, and increases slightly at the later stages.

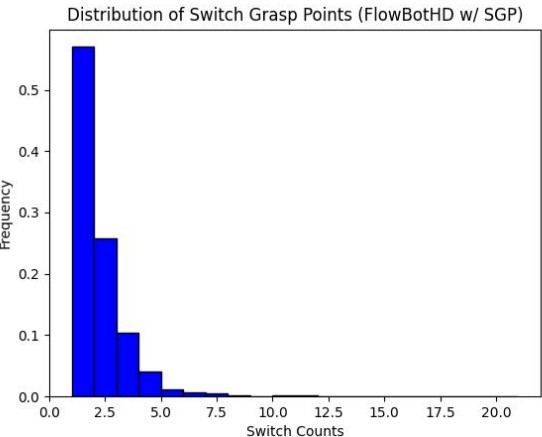

Figure 12: Switch Grasp Count Histogram: we show the distribution of the number of switch grasps across the dataset. About 60% of objects only require one grasp action (the first step). For the other 40 %, most of them require only $< 5$ extra switches of the grasp point.

We performed an additional analysis to calculate the number of grasp actions each object needs. By average, our method needs 1.76 times grasp actions including the initial grasp, so on average each object requires 0.76 switch-grasps. The results show that our method does not require a lot of switching and still leads to improved performance over the baseline.

We made 2 supplementary plots to visualize the frequency of switching grasp points in our simulation experiments. Figure 11 shows the switch grasp point rate per step. We can see that the switch grasp point rate gets lower among the first few steps - the consistency increases with multi-modality decreased. But it increases slightly at the later stages - possibly meaning that occlusions could happen and destabilize the grasp point predictions. Figure 12 shows the distribution of the number of switch grasps across the dataset. We can see that about 60% of objects only require one grasp action (the first step). And for the other 40%, most of them require only <5 extra switches of the grasp point.

### E.3 Step Number Analysis

During a roll-out with limited number of steps (in our case, 30 steps), failures can happen either because of bad prediction quality that will never succeed, or a less efficient strategy or grasp point that might eventually succeed given enough steps. To further analyze the time dependency of the success rate we measured, we made a plot of the opening success rate at each time step.

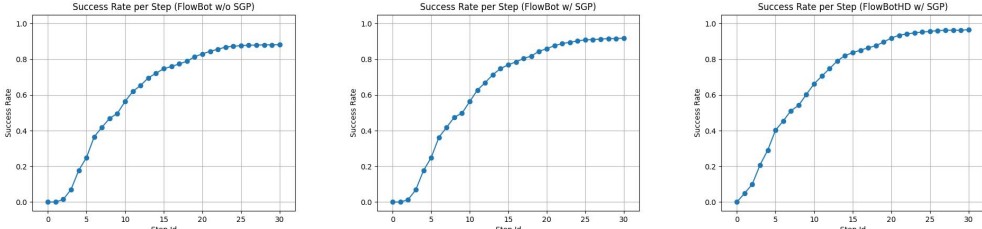

Figure 13: Success Rate over Timestep: We can see that both baseline with and without switch grasp policy (SGP) and FlowBotHD (ours) have nearly reached a plateau after 25 steps. We infer that the performance is not much limited by the 30 step timesteps constraint.

We can see in Figure 13 that both FlowBot3D (baseline) with and without switch grasp policy (SGP) and FlowBotHD (ours) have nearly reached a plateau after 25 steps, indicating that further timesteps after 25 would likely have limited performance increase. In our paper, we allow each method 30 timesteps to open the object before we compute the success rate for our analysis. Therefore we infer that most of the remaining failures are the result of incorrect predictions and the performance is not much limited by the 30 timestep constraint.

## F  Real-world Equipment

### F.1  Hardware

In the real-world demos, we deploy our model on a Franka Emika Panda Robot. We obtain point cloud input data from an Azure Kinect Depth Camera. The robot's end effector is a Schmalz Cobot Pump with a suction cup 3 cm wide in diameter.

### F.2  Workspace

We constructed our workspace (Fig 14) in a 1.3 m by 1.3 m by 1.1 m space. The Azure Kinect Camera was placed so that it pointed toward the center of the workspace at an angle that allowed the door to be seen clearly. To avoid collisions with the table when motion planning, we added a box representing the table top using MoveIt's collision box construction tool.

### F.3  Foreground Segmentation

To isolate the objects in the point cloud, we had to programmatically segment the table, background, and robot points.

To segment out the table and background, we threshold the x, y, and z values of the points to effectively crop all points outside of the box where the object would be. For example, all points with a z value below 0.02 were removed, as that marks the top of the table.

The robot points are filtered out in real-time using a ROS package called Robot Body Filter that takes in the robot's URDF file. The filter assumes perfect calibration of the camera, which leads to some trailing points from the robot occasionally remaining in the scene. We filter out those points using outlier removal since they are sparser than the object's points.

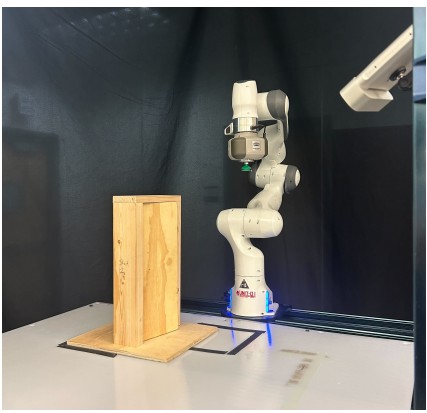

Figure 14: Real World Workspace: This picture depicts the ambiguous door, Franka robot, and Azure Kinect camera.

## F.4 Robot Control Paradigm

The robot is controlled using position control by inputting the desired end-effector pose to MoveIt, which would then motion plan and execute the path.

To make contact, the robot aligns itself with the chosen grasp point and estimated flow direction, then moves to a point 10 cm in the flow direction away. Then, a function is called to start the pump, and the robot moves in 2 cm increments towards the contact point. Another function detects when a vacuum has successfully been established, meaning the robot can progress to opening the object.

For each step, the robot moves 5 cm in the direction of the respective predicted flow vector. After the robot attempts to execute the initial step, we use the switch grasp policy to determine if it should attempt to remake contact. To remake contact, the robot moves back to its initial starting pose, then repeats the process described above. The robot can remake contact up to 5 times.

## F.5 Custom Multi-modal Door

We custom built a multi-modal door out of plywood, with four pairs of hinges. This allows for it to be configured to open in all four of the possibilities of a standard door (forward to the right, forward to the left, backward to the right, and backward to the left). The door's current configuration is determined by two thin allen key-shaped metal rods, which connect a pair of hinges. Those hinges are then effectively in use and determine how the door opens. The door's frame is 25 cm by 8.5 cm by 43.8 cm, with a 31.75 cm by 36 cm by 1.25 cm stabilizing base. The door itself is 16.83 cm by 1.75 cm by 43.5 cm, with nothing on its front or back face to differentiate which way it opens. To ensure that it fits within our workspace, we built it to be much smaller than a normal door, then scaled the point cloud up by three before passing it into the model.

Given the custom made door, our model has the ability to make multi-modal predictions and open the door in all 4 different ways: pull left, pull right, push left and push right. It also has the ability to actively switch grasp point when the initial prompts fail to align with the current configuration. In all the experiments, the action of 'push' is executed as pulling from the back.

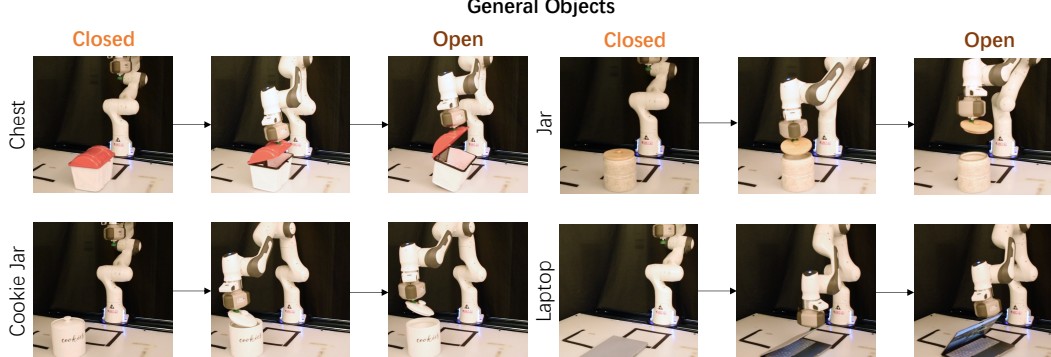

Figure 15: Demonstration of FlowBotHD opening general articulated objects in real-world.

