# OpenReview forum: "FlowBotHD: History-Aware Diffuser Handling Ambiguities in Articulated Objects Manipulation"
_robot-learning.org/CoRL/2024/Conference — CoRL 2024_

### Official Review · Reviewer_aMTY · 2024-07-19
**Solid paper, even if slightly incremental**

**Originality:** 2
**Technical Quality:** 4
**Clarity Of Presentation:** 4
**Potential Impact:** 2
**Recommendation:** 3
**Confidence:** 4

**Review:**

## Strengths

The paper is well written and well presented. The contributions are clearly stated, and the experiments support the claims made by the authors. The proposed approach does, in fact, handle the ambiguities which caused failure cases in the original FlowBot3D.

While reading the paper, I found the insights about adding actions to the manipulation policy interesting. The fact that the new actions are tested with the previous method as well is a welcome baseline comparison, and rounds out the suite of experiments.

## Weaknesses

The central weakness of this paper, in my opinion, is the scope of the contributions. While the proposed approach solves a set of problems from the baseline approach, it is just that - a solution to a limitation of a previous approach, and therefore has limited novelty. A resolution of this weakness could comprise of exploring additional solutions to the ambiguity problem, or on the other hand, more convincing results for the proposed approach (see question 3 below).

Another minor drawback of the paper is that the real-world experiments are lacking; it would be great to see them extended for future versions of the paper, so that they more closely match the set of simulated experiments.

Some other concerns are raised in the questions section below.

**Quality Of The Limitations Section:**

3

**Questions For Rebuttal:**

1. Line 189 states “it might be unclear which direction these objects open”: is there a quantification on how many of the objects in the dataset are actually ambiguous in that sense? While it is clear that doors, for example, are more ambiguous than laptops, even inside the door class I assume some have visual tells such as handles. This quantification would be interesting both in advance (when surveying the dataset), and when learned by the model, i.e., how often does the model predict objects to have multi-modal flows.

2. Is a rollout limited in the amount of time steps? One might expect that even less efficient grasp points to succeed given enough time, as long as they are using the correct flow direction. Do failures mostly happen because of time constraints or because of choosing the wrong grasp points?

3. In the results table, it seems like there are cases where the original FlowBot3D outperforms FlowBotHD. To the best of my understanding, it seems like FlowBotHD extends the previous approach, using a more expressive distribution. Especially with the consistency check heuristic, it is hard to understand why there would be cases where the proposed approach is outperformed. Could the authors share examples for some failure cases where FlowBot3D outperforms FlowBotHD? Is there any insight on when and why this happens?

**Robotics Focus:**

4

**Summary Of Paper:**

This paper proposes an extension to FlowBot3D, a method for predicting the movement flow of articulated objects. The extension is aimed at solving multi-modality and occlusion ambiguities with regards to object movements, and uses diffusion models and history encoding to solve both ambiguities respectively.

**Summary Of Recommendation:**

This paper proposes a valid solution to challenges in a previous approach. While it has some drawbacks, in particular being slightly incremental, I find the contributions good enough to merit acceptance, especially if the other concerns are addressed.

---

### Official Review · Reviewer_niS8 · 2024-07-20
**Manuscript Review**

**Originality:** 4
**Technical Quality:** 4
**Clarity Of Presentation:** 5
**Potential Impact:** 4
**Recommendation:** 3
**Confidence:** 5

**Review:**

The authors present a promising method for handling ambiguities and occlusions while interacting with articulated objects. While the approach demonstrates clear merit, the evaluation strategy needs to include more real-world examples to illustrate the model's effectiveness and maximize the impact of this work.

**Strengths**:
- Addresses a Significant Practical Challenge: The authors tackle the real-world problem of disambiguating articulation directions in articulated objects under challenging conditions, such as occlusions and varying lighting. This research has direct implications for the development of robust action-oriented models and robotic interaction systems in real-world environments.
- Novel and Potentially Impactful Solution: The proposed FlowBotHD approach, utilizing a history-aware diffusion network, represents an innovative solution to address ambiguities in articulation direction. This novel methodology could lead to significant advancements in both perception and manipulation tasks for articulated objects.
- Clear and Comprehensive Presentation: The paper is well-written, with informative figures (notably Figures 2 and 3) and tables that effectively illustrate key concepts. The authors provide clear explanations and anticipate potential reader questions, resulting in a comprehensive presentation of their work.
- Practical Considerations Highlighted: The authors demonstrate a focus on real-world applicability by discussing specific corrections like switch grasp points and consistency checks. This attention to practical implementation details enhances the credibility of the proposed approach.

**Weakness**:
- Limited Real-World Evaluation: While the approach is promising, the current evaluation relies predominantly on synthetic data and simulated environments. A more comprehensive evaluation, incorporating real-world experiments with diverse articulated objects, is necessary to fully validate the effectiveness and generalizability of FlowBotHD.
- The experimental section would benefit from a more in-depth analysis of the results. In particular, the negative impact of the history filter on the performance of certain revolute joint objects (e.g., microwave, locker, washing machine) warrants further investigation and discussion to understand the underlying causes and potential mitigations.

**Miscellaneous**:
- Jain et al. introduced an approach in [1] that directly predicts the network’s uncertainty over the estimated articulation parameters for articulated objects. Consider including the work in the related work section for a more complete literature review on the topic.
- Lines 100-104, please cite the existing methods that suffer from the typical failure modes discussed in the text.
- The justification for the switch grasp point in lines 154-155 could be further strengthened by drawing parallels to human behavior. Humans often intuitively switch grasp points to achieve optimal motion and control when manipulating objects.
- Minor corrections:
  - Line 98 to `manipulation` →`to manipulate`.
  - Line 201 `datasetwith` -> `dataset with`.

*References*:
- [1]  Jain, A., Giguere, S., Lioutikov, R., & Niekum, S. (2022, January). Distributional depth-based estimation of object articulation models. In Conference on Robot Learning (pp. 1611-1621). PMLR.

**Quality Of The Limitations Section:**

2

**Questions For Rebuttal:**

- RGB/RGBD Baselines: Given that RGB features can aid disambiguation in varying lighting conditions, did you consider comparing your method to RGB-only or RGBD-only baselines, such as those in [2, 3, 4]? This could further highlight the value of your approach.
- In-Depth Analysis of Results: In Table 1, the history filter appears to negatively impact performance for some revolute joint objects (e.g., microwave (col. 6), locker (col. 9), washing machine (col. 14)). Could you provide a more detailed analysis in the experimental section, exploring potential reasons for this and discussing any mitigating strategies?
- History Window Length: Why was a history window length of 1 chosen? Would a longer history window potentially improve disambiguation? Did you conduct any ablation studies to determine the optimal window length?
- Frequent Grasp Switching: How does your approach mitigate the issue of frequent grasp switching between time steps? Are there mechanisms in place to ensure stable grasp selection?
- What value of $\delta m$ was used in your experiments?


*References*:
- [2] Mo, K., Guibas, L. J., Mukadam, M., Gupta, A., & Tulsiani, S. (2021). Where2act: From pixels to actions for articulated 3d objects. In Proceedings of the IEEE/CVF International Conference on Computer Vision (pp. 6813-6823).
- [3] Jiang, Z., Hsu, C. C., & Zhu, Y. (2022). Ditto: Building digital twins of articulated objects from interaction. In Proceedings of the IEEE/CVF Conference on Computer Vision and Pattern Recognition (pp. 5616-5626).
- [4] Heppert, N., Irshad, M. Z., Zakharov, S., Liu, K., Ambrus, R. A., Bohg, J., ... & Kollar, T. (2023). Carto: Category and joint agnostic reconstruction of articulated objects. In Proceedings of the IEEE/CVF Conference on Computer Vision and Pattern Recognition (pp. 21201-21210).

**Robotics Focus:**

3

**Summary Of Paper:**

Interacting with articulated objects in real-world environments presents challenges like ambiguous manipulation directions, multiple optimal actions, and occlusions. To address this, the authors introduce FlowBotHD, a history-aware diffusion network that models the multi-modal distribution of object articulation. By incorporating historical context and consistency checks, FlowBotHD disambiguates actions and makes stable predictions even under occlusions. The authors demonstrate their approach's effectiveness by conducting experiments on multiple simulated objects and a real-world door-opening task.

**Summary Of Recommendation:**

The authors' proposed method shows potential for effectively handling ambiguities and occlusions when interacting with articulated objects. However, to fully substantiate the claims and demonstrate the broader applicability of the approach, it is essential to provide more comprehensive evaluation with real-world examples.  I am inclined to recommend this paper for acceptance, contingent upon the authors addressing these concerns through further experimentation and analysis.

---

### Official Review · Reviewer_LDDR · 2024-07-21

**Originality:** 3
**Technical Quality:** 3
**Clarity Of Presentation:** 4
**Potential Impact:** 3
**Recommendation:** 3
**Confidence:** 3

**Review:**

**Key Contributions:**

1.  The paper presents a history-aware diffusion network that addresses these ambiguities by modeling the multi-modal nature of the problem and using historical data to make stable predictions even when occlusions occur.
2. The proposed method leverages diffusion techniques to handle the multiple possible ways an object can be manipulated and adjusts predictions based on past interactions.
3. The approach extends prior work on articulated object manipulation (3DAF vectors) which, while effective in simulations, struggled with ambiguous real-world scenarios.

**Strength**:

1. The paper excels by innovatively using diffusion techniques to manage multi-modality in articulated object manipulation, offering multiple potential solutions for uncertain interactions.
2. It is clearly presented, with a thorough and accessible explanation of the methodology and results.
3. it includes extensive experimental validation, demonstrating the method's effectiveness in handling ambiguities and occlusions across various scenarios.

**Quality Of The Limitations Section:**

2

**Questions For Rebuttal:**

Limitations:

1.  while the proposed method shows strong theoretical and simulated results, there is a lack of hardware contribution, and the real-world experiments are somewhat limited in scope.  This absence means the full practical impact and reliability of the method in diverse, real-world settings remain uncertain.

2. The experiments do not include direct robot testing, which could provide deeper insights into the method’s performance in practical robotic applications. Incorporating such robotic experiments could offer valuable validation and could be more appropriate for discussions in other conferences focused on practical robotics and implementation challenges.

**Robotics Focus:**

1

**Summary Of Paper:**

The paper introduces a new method for manipulating articulated objects with inherent ambiguities, such as doors with uncertain opening mechanisms (push, pull, slide) and directions. These ambiguities are compounded by occlusions that obscure the object's shape.

**Summary Of Recommendation:**

Weak accept.

---

### Author Rebuttal · Authors · 2024-08-14

Thanks for all the reviewers' valuable comments and feedback! We have incorporated your suggestions to improve the paper. The Rebuttal_Supplementary.pdf file includes all the figures of the extra experiments and analysis we conducted.

### 1. Real-world experiments

We performed real-world experiments on general objects and specifically our custom made multi-modal door to demonstrate FlowBotHD’s ability to function in real-world scenarios. For the multi-modal door, we demonstrate our model's ability to open the door in all 4 possible modes, and also the ability to adjust to better grasp points with the switch grasp point policy.

Details of the real-world settings can be found in Rebuttal_Supplement.pdf, and videos are shown on our anonymized website:[ https://flowbothd.github.io/](https://flowbothd.github.io/).

### 2. Paper contribution

We propose a history-aware diffusion model to handle ambiguity issues including multi-modality and occlusions, incorporating “switch grasp point”, “consistency check,” and “history filter”, and our ablations show the value of each of these components. The ambiguity issues that we are dealing with would apply to any model and other experimental settings, even though we are only testing these ideas based on FlowBot and articulated objects. Therefore, we hope that our proposed method could offer insights to improve a broader range of methods and tasks.

---

### Decision · Program_Chairs · 2024-09-04

**Decision:**

Accept

**Comment:**

Strengths:
- Clearly presented with extensive experimental validation.
- History aware diffusion networks that offers potential for handling uncertain situations in object manipulation like occlusions.

Weaknesses:
- Lack of real world results and direct evaluation on robots
- Scope of contribution is limited: provides a solution to limitations of prior methods

Reviewer LDDR focused on the lack of real-world experiments and direct robot testing. In response, the authors provided details about their real-world experiments, including the hardware, workspace setup, foreground segmentation process, and robot control paradigm. They also discussed the custom-built multi-modal door used in their experiments.

Reviewer niS8 also pointed out the limited real-world evaluation and suggested including more real-world examples. Additionally, they asked about RGB/RGBD baselines, in-depth analysis of results, history window length, frequent grasp switching, and the value of om. The authors addressed these points by providing further analysis of their results, explaining their choices in the experimental setup, and discussing the challenges of incorporating RGB/RGBD baselines. They also included real-world experiments with a custom-built multi-modal door.

Reviewer aMTY questioned the scope of the contributions, the limited real-world experiments, and specific aspects of the method, such as the quantification of ambiguous objects, rollout time steps, and cases where FlowBot3D outperforms FlowBotHD. The authors responded by providing a more detailed analysis of ambiguities in the dataset, discussing the time dependency of their method, and explaining the reasons behind the occasional outperformance of FlowBot3D. They also extended their real-world experiments to address the reviewer's concerns.